# Integrated action of pheromone signals in promoting courtship behavior in male mice

**Sachiko Haga-Yamanaka[1], Limei Ma[1], Jie He[1†], Qiang Qiu[1], Luke D Lavis[2], Loren L Looger[2], C Ron Yu[1,3]\***

[1]Stowers Institute for Medical Research, Kansas City, United States; [2]Howard Hughes Medical Institute, Janelia Farm Research Campus, Ashburn, United States; [3]Department of Anatomy and Cell Biology, University of Kansas Medical Center, Kansas City, United States

**Abstract** The mammalian vomeronasal organ encodes pheromone information about gender, reproductive status, genetic background and individual differences. It remains unknown how pheromone information interacts to trigger innate behaviors. In this study, we identify vomeronasal receptors responsible for detecting female pheromones. A sub-group of V1re clade members recognizes gender-identifying cues in female urine. Multiple members of the V1rj clade are cognate receptors for urinary estrus signals, as well as for sulfated estrogen (SE) compounds. In both cases, the same cue activates multiple homologous receptors, suggesting redundancy in encoding female pheromone cues. Neither gender-specific cues nor SEs alone are sufficient to promote courtship behavior in male mice, whereas robust courtship behavior can be induced when the two cues are applied together. Thus, integrated action of different female cues is required in pheromone-triggered mating behavior. These results suggest a gating mechanism in the vomeronasal circuit in promoting specific innate behavior.

**\*For correspondence:** cry@stowers.org

**Present address:** [†]Institute of Neuroscience, Chinese Academy of Sciences, Shanghai, China

**Competing interests:** The authors declare that no competing interests exist.

**Reviewing editor**: Jeremy Nathans, Howard Hughes Medical Institute, Johns Hopkins University School of Medicine, United States

## Introduction

Pheromones play an essential role in reproductive behaviors. Sexually mature animals respond to pheromone signals from potential mates with endocrine changes and ritualistic action patterns (**Wyatt, 2003**). In spite of their innate nature, reproductive behaviors are not simple reflexes. They are complex neural processes that are modulated not only by the reproductive status of the animal (e.g., hormonal states, social ranks and past experiences), but also the status of potential mates and environmental factors such as the presence of predators (**Tinbergen, 1951**). A fundamental question in understanding pheromone communication is how the confluence of different signals works to trigger innate behaviors.

Vomeronasal organ (VNO), the major sensory organ responsible for pheromone detection in mammalian species, expresses more than 300 vomeronasal receptors (VRs) (**Dulac and Axel, 1995**; **Herrada and Dulac, 1997**; **Matsunami and Buck, 1997**; **Ryba and Tirindelli, 1997**; **Dulac and Torello, 2003**; **Shi and Zhang, 2007**; **Young and Trask, 2007**; **Liberles et al., 2009**; **Riviere et al., 2009**; **Liberles, 2014**). The vomeronasal sensory neurons (VSNs) encode information about gender, reproductive status, genetic background and individual differences of animals of the same species, as well as that of heterospecifics (**Halem et al., 1999**; **Holy et al., 2000**; **Boschat et al., 2002**; **Wyatt, 2003**; **Leinders-Zufall et al., 2004**; **Kimoto et al., 2005**; **Chamero et al., 2007**; **Kimoto et al., 2007**; **He et al., 2008**; **Haga et al., 2010**; **Papes et al., 2010**; **Isogai et al., 2011**; **Tolokh et al., 2013**). The multitude of cues potentially provides snapshots of other animals and surrounding environments to influence reproductive

**eLife digest** Pheromones are chemicals that are given off by living things and they lead to a range of social responses in others of the same species. These chemical signals, for example, can let an animal know when a suitable mate is near and trigger the release of hormones that encourage the animal to mate.

In mammals, an organ found between the roof of the mouth and the nose detects pheromones. This organ contains more than 300 different receptors for these chemicals. However, only a few of these receptors have been matched with the pheromones that they detect. One example is a chemical released by male mice that interacts with a specific nasal receptor and causes a female mouse to arch her back in a way that signals she is ready to mate.

One reason that more pheromone-receptor pairs are not known is that pheromones are released in very small quantities, which makes them hard to detect. In an effort to identify more pairs, Haga-Yamanaka et al. took tissue slices from the organ that detects pheromones in mice and then looked for cells that responded to the urine of female mice. Two previously unknown pheromone-receptor pairs were found. One helps male mice detect when a female is present, while the other lets him know if she is ready to mate. Together these two chemicals alert a male mouse to a potential mate and cause him to mount her in order to mate. However, neither chemical is able to trigger this male courtship behavior on its own.

The techniques developed by Haga-Yamanaka et al. may, in the future, help identify more pheromone-receptor pairs. The next challenge will be to identify the pathways of nerve cells that integrate the information about pheromones and trigger the courtship behaviors.

behaviors. To date, our understanding of the neural logic that controls these innate behaviors remains limited. It has been shown that individual pheromones can directly trigger distinct innate behaviors. For example, the male-specific exocrine peptide ESP1 induces lordosis in female mice, a stereotypic gesture characterized by the arching of the back with raised rear, to signal receptivity to potential mates (*Kimoto et al., 2005*; *Haga et al., 2010*). ESP1 induces lordosis through its receptor V2Rp5 (Vmn2r116), a member of the V2r family (*Haga et al., 2010*). In this case, the circuit that connects receptor-expressing neurons to brain centers appears to be a labeled line—the activation of a single receptor and its downstream neurons is sufficient to trigger lordosis, and removing the V2Rp5 receptor abolishes both the ESP1-evoked neural activation in the brain and the behavioral output of ESP1 (*Haga et al., 2010*).

On the other hand, there is likely more complex circuit logic than labeled lines in the vertebrate species (*Wyatt, 2003*). In contrast to the myriad sensory cues that induce behavioral responses and the large number of putative receptors, there is a limited repertoire of highly stereotyped behavioral outputs resulting from VNO activation. In genetic and surgical disruptions of VNO function that lead to the alterations of innate behaviors, it is often observed that the onset and frequency of specific behaviors, but not the behavioral patterns per se, are altered. This revelation indicates that the signals processed by distinct VR circuits are likely integrated in brain networks to induce a behavior selected from a fixed set of behavioral patterns. Thus, the neural circuits that process pheromone information must have an intrinsic logic in integrating external and internal signals to produce coordinated behaviors in order to maximize reproductive fitness. This logic has not been clearly illustrated. Given that only a few receptors among the more than 300 members of the vomeronasal receptor families have been paired with ligands and assigned with functions (*Boschat et al., 2002*; *Shi and Zhang, 2007*; *Young and Trask, 2007*; *Haga et al., 2010*; *Liberles, 2014*), identification of receptors responding to specific pheromone signals should provide us with insights into the logic of information processing in the vomeronasal system.

To date, despite advances in identifying pheromones (*Kimoto et al., 2005*; *Chamero et al., 2007*; *Haga et al., 2010*; *Roberts et al., 2010*; *Ferrero et al., 2013*), the receptors for most pheromones have not been identified. In particular, the receptors conveying sex information remain elusive. We seek to identify VRs that convey information about sex and reproductive status of the female mice. In a previous study, we developed transgenic mice expressing the genetically encoded $Ca^{2+}$ sensor GCaMP2 in the VNO neurons (*He et al., 2008*). By profiling the calcium response of individual VNO neurons in slice preparations, we were able to identify neurons that respond to gender- or estrus-specific

urinary stimuli (*He et al., 2008*). These cells likely convey information regarding gender identity and reproductive status of the emitter. In this study, we combine this approach with single-cell degenerate reverse transcription-PCR to clone receptor genes from neurons with specific response profiles. This novel approach allows us to identify vomeronasal receptors that detect specific signals whether or not the ligands are known. The identification of the receptors facilitated the purification of a urinary fraction that activates those receptors. The urinary signals act synergistically to induce robust mounting behavior in male mice. Our study suggests two distinct operating principles in pheromone-triggered mating behaviors.

## Results

### Sulfated estrogens mimic the activity of estrus signal in urine

Male mice exhibit robust mounting behaviors towards sexually mature females in the presence of estrus female urine (*Dixon and Mackintosh, 1975*; *Ingersoll and Weinhold, 1987*). In a previous study, we identified two populations of vomeronasal sensory neurons that respond to gender and estrus status cues from female mouse urine, respectively (*He et al., 2008*). To examine whether these cues communicate sexual receptivity to males, we painted ovariectomized females, which are not influenced by endogenous hormonal and pheromonal fluctuations, with female mouse urine collected during the estrus phase (estrus urine, or EU; 'Materials and methods') or non-estrus phase (non-estrus urine, or NEU), and exposed them to sexually naïve males. Although males showed baseline mounting behavior toward ovariectomized females painted with vehicle or NEU (*Figure 1A*), the number of mounts and mounting duration increased significantly when the females were painted with EU (*Figure 1A*). Thus, estrous female urine contains cues that promote mounting behavior in male mice.

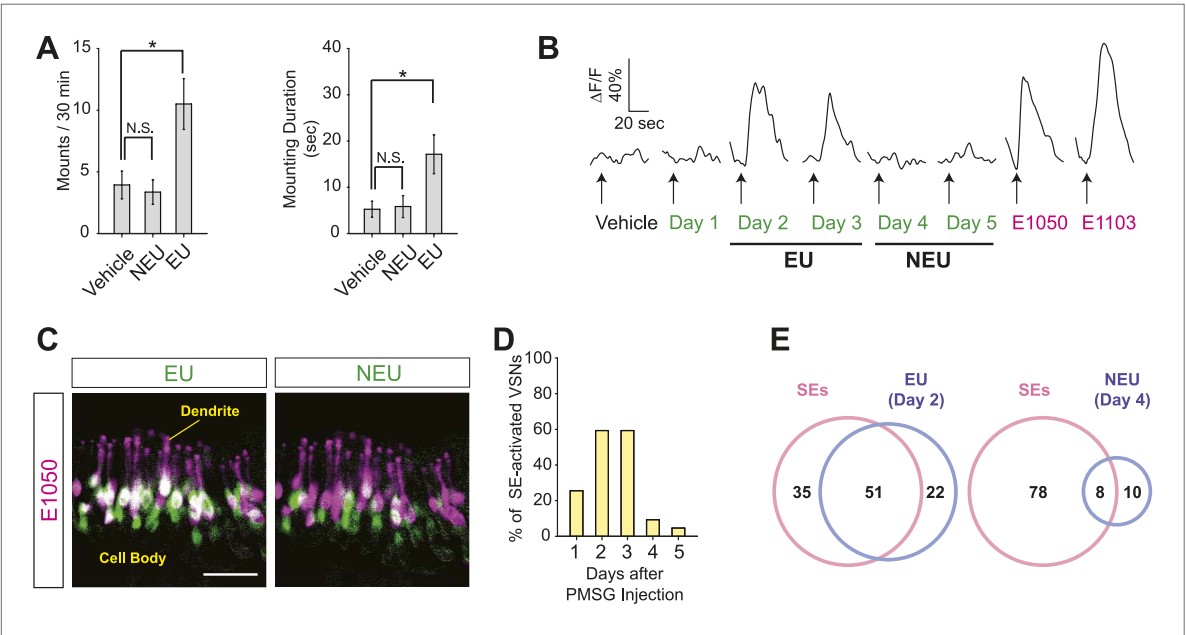

**Figure 1**. Sulfated estrogens mimic the activity of estrus signal in urine. (**A**) The number (left) and duration (right) of mounting behavior of sexually naïve males toward ovariectomized females painted with vehicle (n = 16), non-estrus urine (NEU; n = 14) or estrus urine (EU; n = 22). EU and NEU were collected 2 and 4 days, respectively, after PMSG injection to induce estrus. Error bars, SEM; *p<0.05 (Mann–Whitney test). (**B**) Traces showing GCaMP2 responses of a representative cell to urine samples collected from females 1 to 5 days after PMSG injection and SEs. Arrows indicate the onset of stimulus delivery. (**C**) Representative images of the VNO slice response pattern to E1050 (magenta) and urine (green). Scale bar, 50 μm. (**D**) Bar graph showing the percentage of SE-activated VSNs (n = 86; 3 slices) that are also activated by female urine samples. (**E**) Venn diagrams showing the overlap between VSNs responding to SEs and those activated by EU or NEU.

The following figure supplements are available for figure 1:

**Figure supplement 1**. Activation of the VSNs by female mouse urine.

**Figure supplement 2**. Activation of the VSNs by sulfated estrogen E1050 and E1103.

What are the signals that promote male courtship behavior? Sulfated steroids, including sulfated estrogens (SEs), have been shown to activate VSNs (*Nodari et al., 2008*; *Meeks et al., 2010*). Estrus is marked by a surge in circulating estrogen, which can be converted by specific enzymes to sulfated or glucuronidated estrogens (*Blair, 2010*) that are soluble and excreted in urine (*Shackleton, 1986*). It is plausible that a rise in the level of modified estrogen in female urine during estrus serves as estrus signal. We therefore examined whether SEs activated the population of VSNs specifically tuned to estrus signals (*He et al., 2008*).

We profiled the calcium response of individual VSNs using VNO slices prepared from transgenic mice expressing the genetically encoded Ca$^{2+}$ sensor GCaMP2 (*He et al., 2008*; *Ma et al., 2011*). A population of VSNs responded robustly to EU but not NEU (*Figure 1B*, *Figure 1—figure supplement 1*). Two SEs, 1, 3, 5(10)-estratrien-3, 17β-diol disulfate (E1050) and 1, 3, 5(10)-estratrien-3, 17β-diol 17-sulfate (E1103), exhibited overlapping response patterns (*Figure 1—figure supplement 2A,B*). Remarkably, the majority of neurons activated by SEs were also activated by EU (*Figure 1B–E*, *Figure 1—figure supplement 2C*). EU activated >60% of SE-responding VSNs while NEU activated less than 10% (*Figure 1D,E*), suggesting that SE-responsive cells convey estrus information.

## VSNs responding to sulfated estrogens express V1rj receptors

We next aimed to identify the VRs responsible for detecting estrus signals. The VNO expresses more than 300 VRs, including the V1r, V2r and FPR families of genes (*Liberles, 2014*; *Shi and Zhang, 2007*; *Young and Trask, 2007*; *Dulac and Torello, 2003*; *Dulac and Axel, 1995*; *Herrada and Dulac, 1997*; *Matsunami and Buck, 1997*; *Ryba and Tirindelli, 1997*; *Riviere et al., 2009*; *Liberles et al., 2009*). In order to clone the specific receptors, we identified VSNs that showed robust responses to either E1050 or E1103 and captured their cytoplasmic contents using micro-capillaries (*Figure 2A*). The lysed cytoplasmic contents were used as substrates for reverse transcription and cDNA amplification (*Kurimoto et al., 2007*), followed by PCR using degenerate primers designed against the V1r and V2r family members (*Figure 2B*; *Supplementary file 1*). The 24 pairs of primers (17 for V1rs and 7 for V2rs) were designed to capture all known V1r and V2r genes. The primers were tested by using pooled VNO mRNA as well as single cell samples, and were confirmed to have the broad coverage (*Supplementary file 2*).

From cells responding to either E1050 or E1103, we repeatedly identified members of the V1rj clade of receptors (*Figure 2C*). We isolated 22 VSNs that responded to the two compounds, 15 of which yielded PCR products amplified specifically by a subset of degenerate primers. From these 15 cells, we identified V1rj2 (Vmn1r89) and V1rj3 (Vmn1r85) from 10 and 3 cells, respectively. Vmn1r86 (The sequence is identical to V1rj1 (http://www.ncbi.nlm.nih.gov/nuccore/AY044668.1), but V1rj1 is not annotated in BLAT or Ensembl databases, nor was it linked to Vmn1r86.) was found from one cell. The V1rj members are located on mouse Chromosome 7 (Chr.7) and cluster into a single group in phylogenetic analysis (*Figure 2D*). They share 55–86% amino acid identity and 81–95% similarity (*Figure 2—figure supplement 1*). Although other receptor transcripts were found in some VSNs (*Figure 2C*), none of them was reproducibly identified from other cells. Those genes were therefore considered as experimental contaminations.

## V1rj receptors selectively respond to sulfated estrogens and estrus urinary cues

We next tested whether the V1rj receptors convey the specificity of response to estrus cues. In the absence of a robust heterologous expression system of VRs to examine their functions, we took a transgenic approach to ectopically express either V1rj2 or V1rj3 receptor in VSNs using the binary tetracycline-transactivator (tTA) system (*Figure 3A,B*) (*Nguyen et al., 2007*; *Fleischmann et al., 2008*). Mice carrying bicistronic tetO-V1rj2-IRES-tdTomato or tetO-V1rj3-IRES-tdTomato allele (*Figure 3A*, iii) were mated to compound lines in which tTA expression is driven by the promoters of G protein γ-subunit 8 (Gγ8-tTA, *Figure 3A*, i) and olfactory marker protein (OMP-IRES-tTA or OIVT; *Figure 3A*, ii). Neurons expressing ectopic V1rj2 or V1rj3 gene also express the fluorescent protein tdTomato (*Figure 3B*). We further crossed the animals with those carrying the tetO-GCaMP2 allele (*Figure 3A*, iv). The progeny possessing these 4 transgenes had large number of VSNs expressing both GCaMP2 and the V1r-tdTomato transgenes (*Figure 3B*), which allowed us to analyze neural responses by monitoring GCaMP2 signal in V1rj2/3-tdTomato-expressing neurons (*Figure 3C*).

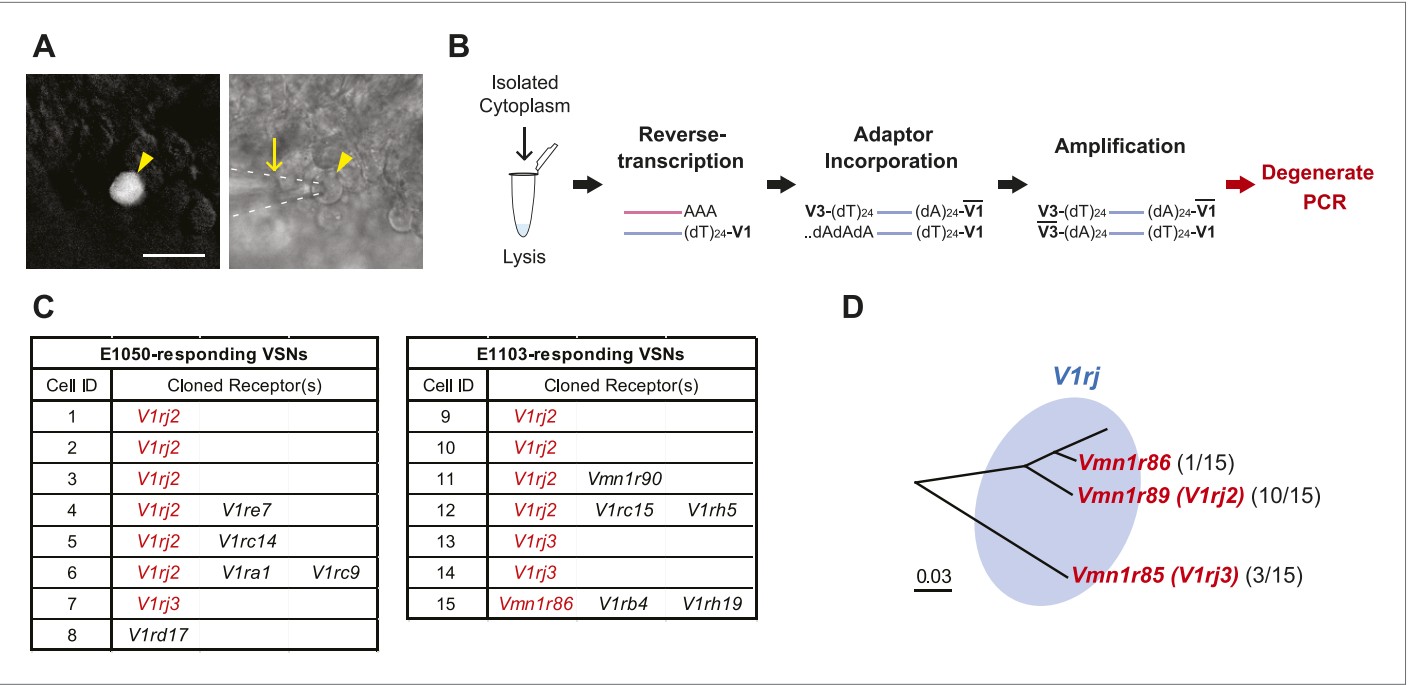

**Figure 2**. VSNs responding to estrus signal express V1rj receptors. (**A**) Representative images of the single-cell isolation procedure. Arrowheads indicate a responding cell illuminated under fluorescence (left), which is aspirated into a micro-capillary (arrow) under bright field illumination (right). Scale bar, 20 μm. (**B**) Schematic illustration of single-cell degenerate RT-PCR procedure. (**C**) Tables showing the receptor genes cloned from individual E1050-responding (left) and E1103-responding VSNs (right). Members of V1rj clade are indicated in red. (**D**) Enlarged view of phylogenetic tree showing the V1rj clade. Receptors cloned from SE-responding VSNs are indicated in red. Numbers in parentheses indicate the number of the VSNs that expressed the receptor out of the total number of SE-responding VSNs profiled.

The following figure supplements are available for figure 2:

**Figure supplement 1**. Members of V1rj group receptors are homologous to each other.

SEs at 100 nM concentration activated ~10% of VSNs in control VNO slices (*Figure 3C*, *Figure 3—figure supplement 1*). Significantly higher number of cells responded to 100 nM SEs in the V1rj2- and V1rj3-tdTomato-expressing VNOs (*Figure 3C*, *Figure 3—figure supplement 1*). The V1rj2 and V1rj3 expressing VSNs showed dose-dependent activation, responding to the SEs at concentrations as low as 0.1 nM (*Figure 3D,E*). The ligand specificity of these receptors were tested by using a variety of sulfated steroids (*Supplementary file 3*), including sulfated androgens (e.g., A2460, A6940 and A7864), corticosterones (e.g., Q1570 and Q3910) and progesterones (e.g., P8168 and P2135) and other estrogen compounds (*Figure 3F,G*). Interestingly, while V1rj2 was narrowly-tuned to sulfated estrogens, it also responded to A7864, which had been previously shown to activate V1rj2 (*Isogai et al., 2011*). V1rj3, on the other hand, showed more restricted tuning and was only activated by E1050 and E1103.

Importantly, EU strongly activated both V1rj2- and V1rj3-expressing VSNs while NEU activated them to a much lesser extent (*Figure 4*). V1rj3-expressing VSNs were also more strongly activated by urine samples from estrous females under natural estrus cycle (*Figure 4—figure supplement 1*). These results suggest that both V1rj2 and V1rj3 are receptors for the urinary cues enriched in EU that convey information about the estrus status of the female animals.

## V1re-Chr.7 group receptors recognize female-specific gender signals

Our studies suggest that there are at least two sets of female pheromone cues in urine. Estrus signals fluctuate with reproductive cycles. In contrast, cues that activate the female urine specific cells (FUSCs) are present regardless of genetic background or estrus status of the donors (*He et al., 2008*). Activation of FUSCs may convey female specific gender-identifying information. To identify VRs expressed in FUSCs, we profiled VSN response to multiple urine samples from either sex (*Figure 5A*), and isolated

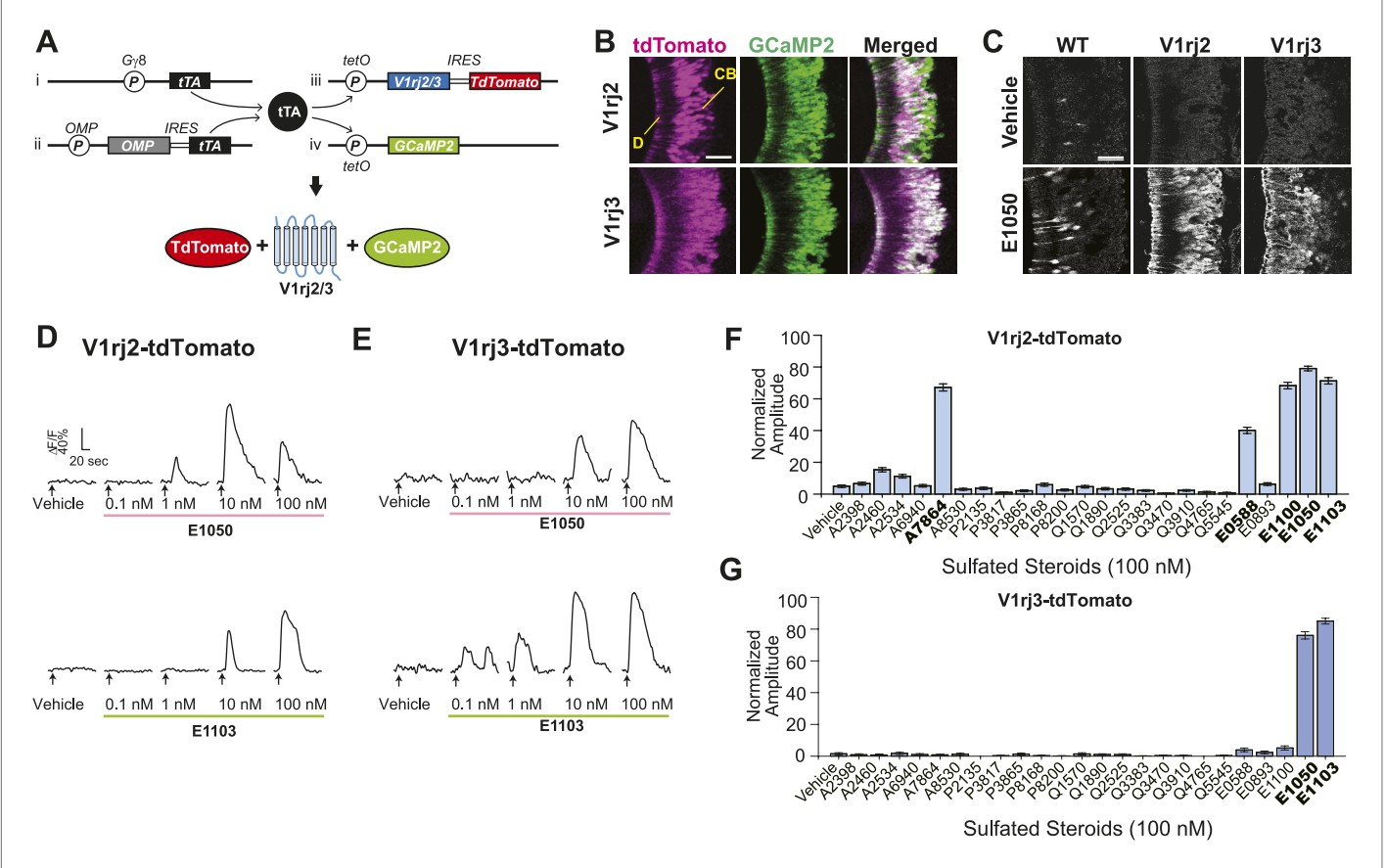

**Figure 3**. V1rj receptors selectively respond to sulfated estrogens. (**A**) Schematic illustration of transgenic alleles that induce ectopic V1rj2- or V1rj3-expression in the VNO: (i) Gγ8-tTA allele drives tTA expression in immature VSNs; (ii) Knock-in OMP-IRES-tTA (OIVT) allele drives tTA expression in mature VSNs; (iii) tetO-V1rj2/3-tdTomato allele that allows bicistronic expression of V1rj2 or V1rj3 with tdTomato; (iv) tetO-GCaMP2 allele. (**B**) Representative images of VNO slices from Gγ8-tTA;OIVT;tetO-V1rj2-tdTomato;tetO-GCaMP2 (top) and Gγ8-tTA;OIVT;tetO-V1rj3-tdTomato;tetO-GCaMP2 (bottom) mice. D: dendrite; CB: cell body. Scale bar, 50 μm. (**C**) Representative images of GCaMP2 VNO slices from the control (left), V1rj2- (middle) and V1rj3-tdTomato-expressing (right) mice responding to vehicle (top) or 100 nM E1050 (bottom). Scale bar, 50 μm. (**D** and **E**) Traces showing GCaMP2 responses of two representative V1rj2 (**D**) or V1rj3 (**E**). Response of tdTomato-expressing cells to different concentrations of E1050 (top) and E1103 (bottom). Arrows indicate the onset of stimulus delivery. (**F** and **G**) Bar graph showing the normalized response amplitude of V1rj2 (**F**; n = 247) or V1rj3 (**G**; n = 207) VSNs to a set of sulfated steroid compounds. Error bars, SEM.

The following figure supplements are available for figure 3:

**Figure supplement 1**. E1103 activates V1rj2/3-expressing VSNs.

10 candidates for degenerate RT-PCR analysis. 7 out of the 10 FUSCs were found to express receptors belonging to the V1re clade (*Figure 5B*).

The V1re clade members are found on both Chr.7 and Chr.17. Strikingly, the receptor genes identified from the FUSCs were all located on Chr.7 (referred to as V1re-Chr.7 receptors; *Figure 5B*). Despite their sequence similarity, none of V1re members located on Chr.17 was found in the FUSCs, whereas 5 out of 7 V1re-Chr.7 receptors were cloned from these cells. The cloned V1re-Chr.7 receptors share 82–96% amino acid sequence similarity and 55–89% identity (*Figure 5—figure supplement 1*). We reasoned that it was highly improbable that receptor genes belonging to a single clade are cloned from 70% of the cells by random chance. Thus, V1re-Chr.7 receptors were considered good candidate receptors for female gender-specific cues.

To characterize the response properties of the V1re-Chr.7 receptors, we generated tetO-V1re9-IRES-tdTomato and tetO-V1re12-IRES-tdTomato lines and crossed them to lines containing the OMP-IRES-tTA, Gγ8-tTA and tetO-GCaMP2 alleles (*Figure 5—figure supplement 2*). When urine samples

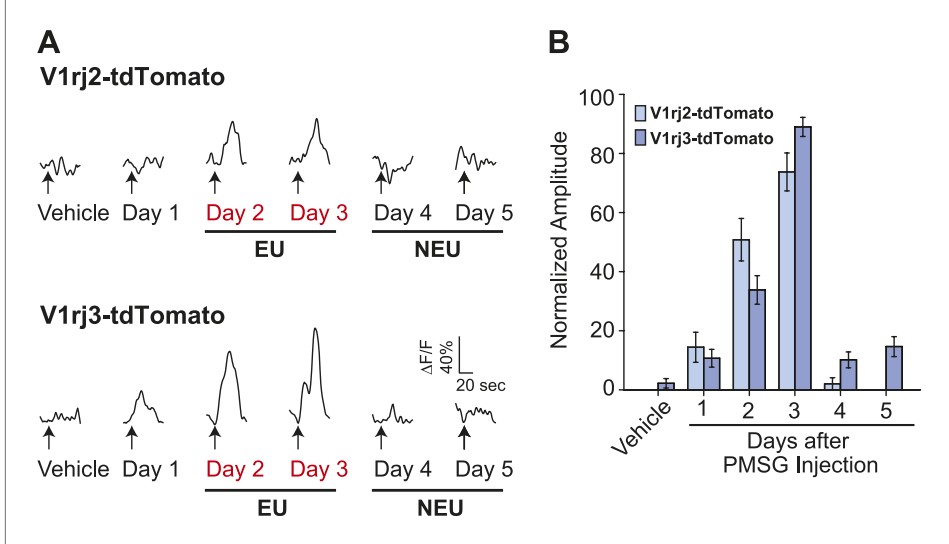

**Figure 4**. V1rj receptors selectively respond to estrus urinary cues. (**A**) Traces showing GCaMP2 responses of a representative V1rj2-tdTomato-expressing cell (top) and a V1rj3-tdTomato-expressing cell (bottom) to urine samples collected from females 1 to 5 days after PMSG injection. (**B**) Bar graph showing normalized response amplitude of V1rj2 (n = 43) or V1rj3 (n = 77) VSNs labeled by both GCaMP2 and tdTomato. Error bars: SEM.
The following figure supplements are available for figure 4:

**Figure supplement 1**. Urine from estrous females in natural estrus cycle activates V1rj3-expressing VSNs.

from males and females of several strains were applied to slices prepared from mice carrying all four alleles, tdTomato-expressing cells were activated by all of the female but not the male samples (*Figure 5C,D*). Importantly, tdTomato-expressing cells were activated by the urine of females regardless of their estrous conditions (*Figure 5E,F*). Urine from ovariectomized females can also activate V1re9 and V1re12-expressing cells (*Figure 5—figure supplement 3*). However, urine from ovariectomized females was less potent in activating these neurons, especially the V1re9-expressing cells. These observations suggested that the expression of female-specific gender cue(s) was likely regulated by the ovaries. Taken together, these results demonstrate that V1re-Chr.7 receptors are activated by gender-specific cues from females.

## Detection of female cues in male and female VNO

Pheromone cues induce distinct behaviors in male and female animals. The identification of receptors for the female cues provides an opportunity to examine sexual dimorphism in detecting the cues. We performed double in situ hybridization using riboprobes against V1re9, e12, j2 and j3 receptor transcripts. We did not find obvious differences between male and female VNOs in the numbers of cells expressing the receptors (*Figure 6A*). Consistent with this finding, the patterns of response to E1050 or to EU were similar between VNOs obtained from male or female mice (*Figure 6B*). Thus, there is no obvious sexual dimorphism in detecting the female cues in the VNO.

## Purification of female-specific gender cues

VSNs are extremely sensitive to their ligands; known ligands activate the VSNs at nanomolar or sub-nanomolar concentrations (*Leinders-Zufall et al., 2000, 2004*; *Kimoto et al., 2005*; *He et al., 2010*). The amount of putative pheromones in urine is likely to be very small, making it difficult to identify them. Taking advantage of the VNOs ectopically expressing V1re9 and V1re12, we performed imaging experiments to screen for urinary fractions that activate these receptors. We first identified urine fractions that retained VNO-stimulating activities (*Figure 7—figure supplement 1*) and subsequently screened high performance liquid chromatography (HPLC) fractions separated using a C18 column with methanol gradient (*Figure 7A*). One of the fractions, T16 (*Figure 7A*), strongly activated the V1re9 and V1re12 cells but not the V1rj2 or V1rj3 neurons (*Figure 7B,C*), suggesting that T16 contains the female gender-specific cue(s).

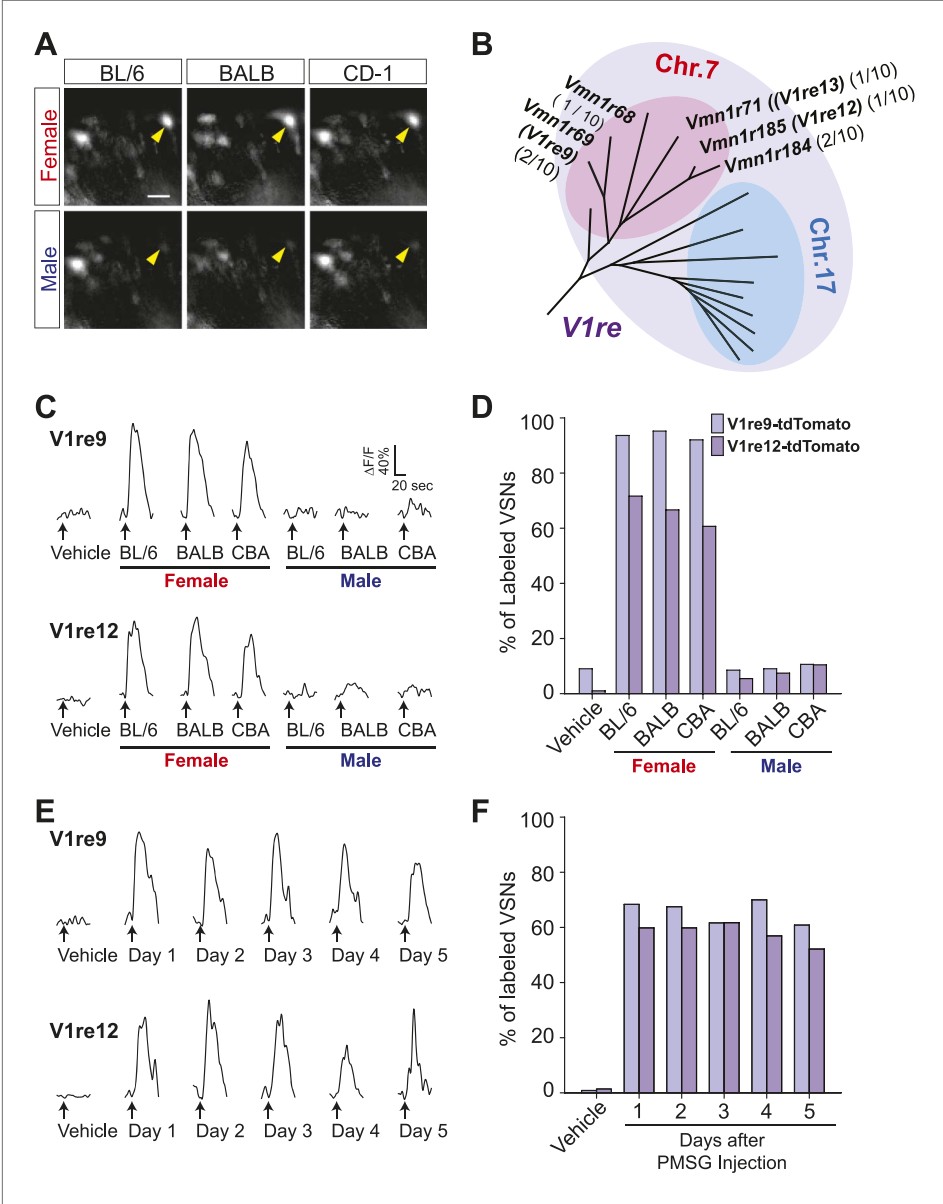

**Figure 5**. V1re-Chr.7 group receptors recognize female-specific gender signals. (**A**) Representative images of VSNs responding to urine samples from either females (top) or males (bottom) of different mouse strains. White arrow-heads indicate a Female Urine Specific Cell (FUSC). Scale bar, 20 μm. (**B**) Enlarged view of the V1re clade of the phylogenetic tree of the V1r family. Receptors located on Chr.7 and Chr.17 are circled in pink and light blue, respectively. Receptors cloned from FUSCs are indicated in bold font. The numbers in parentheses indicate the number of the VSNs that express the receptor over the total numbers of FUSCs examined. (**C**) Traces showing the GCaMP2 responses of a representative V1re9- (top) and a V1re12-tdTomato-expressing cell (bottom) to female and male urine samples from multiple mouse strains. (**D**) Bar graph showing the percentage of V1re9 (n = 195) or V1re12 (n = 201) VSNs responding to urine samples from various strains of males and females. (**E**) Traces showing the GCaMP2 responses of a representative V1re9- (top) and a V1re12-tdTomato-expressing cell (bottom) to urine samples from females 1 to 5 days after PMSG injection. (**F**) Bar graph showing the percentage of V1re9 (n = 120) or V1re12 (n = 209) VSNs responding to female urine samples.

The following figure supplements are available for figure 5:

**Figure supplement 1**. Members of V1re-Chr.7 group receptors are homologous to each other.

*Figure 5. Continued on next page*

*Figure 5. Continued*

**Figure supplement 2**. V1re9/12-expressing VSNs express GCaMP2.

**Figure supplement 3**. Urine sample from ovariectomized females activates V1re9/e12-expressing cells.

## Sulfated estrogen and female-specific gender cues do not activate the main olfactory system

Some putative pheromone cues that activate the VNO have been found to activate the main olfactory system, suggesting that the information may be processed by both systems. We tested whether SEs and the T16 fraction activate the main olfactory epithelium (MOE). We conducted electro-olfactogram (EOG) recordings from the MOE. As a control, we used 2-heptanone, which had been shown to activate neurons of the VNO and MOE. 2-heptanone elicited strong EOG responses (*Figure 8A,B*). In contrast, SEs at 100 nM and 10 µM elicited only background level signals (*Figure 8A,B*). Similarly, neither pure nor diluted T16 fractions elicited EOG responses (*Figure 8C*). Thus, these cues do not activate the MOE, demonstrating that the information is not processed by the main olfactory system.

## Combined female and estrus cues are sufficient to promote mounting behavior

We next tested whether ligands that activate the V1rj and V1re-Chr7 receptors could stimulate male mating behaviors. We painted E1050 and E1103 on ovariectomized females before exposing them to sexually-naïve males, but did not observe increased mounting activities toward those females. The number of mounts and mounting duration were similar to those induced by vehicle or NEU treatment (*Figure 9A*, *Figure 9—figure supplement 1*; also see *Figure 1A*). Interestingly, however, supplementing NEU with SEs induced more robust mounting (*Figure 9—figure supplement 1*), suggesting that additional cues besides the SEs were needed to elicit mounting behavior in males. The additional cue is likely female-specific and is present in both NEU and EU, making T16 fraction a good candidate for such a cue. Indeed, mixing T16 fraction with SEs induced mounting activities as robust as that induced by EU (*Figure 9A*), whereas painting T16 fraction alone did not elicit increased mounting by males (*Figure 9A*). Thus, individually the female pheromone cues are not sufficient to induce mounting. The synergic action of both cues is required to promote male mounting behavior.

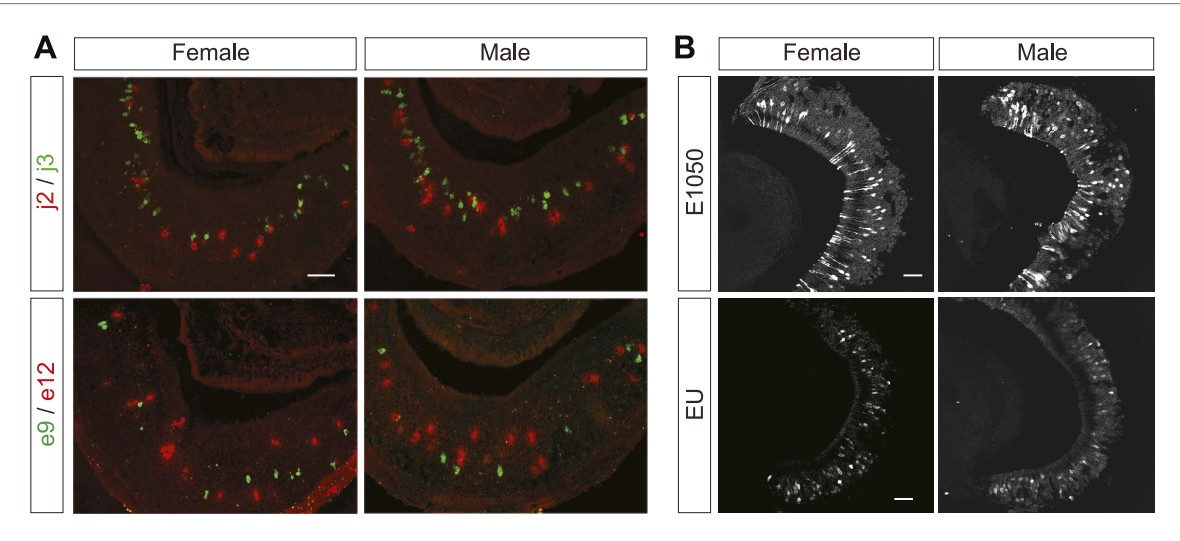

**Figure 6**. The female cues are detected in male and female VNOs. (**A**) Double in situ hybridization of VNO slices. Top row: confocal images showing cells expressing V1rj2 (red) and V1rj3 (green) in VNO sections obtained from female (left) and male (right) mice. Bottom row: confocal images showing cells labeled by V1re9 (green) and V1re12 (red) probes in VNO sections from female (left) and male (right) mice. (**B**) Representative images of GCaMP2 VNO slices from female (left) and male (right) mice responding to 100 nM E1050 (top) and estrus urine (bottom). Scale bars, 50 µm.

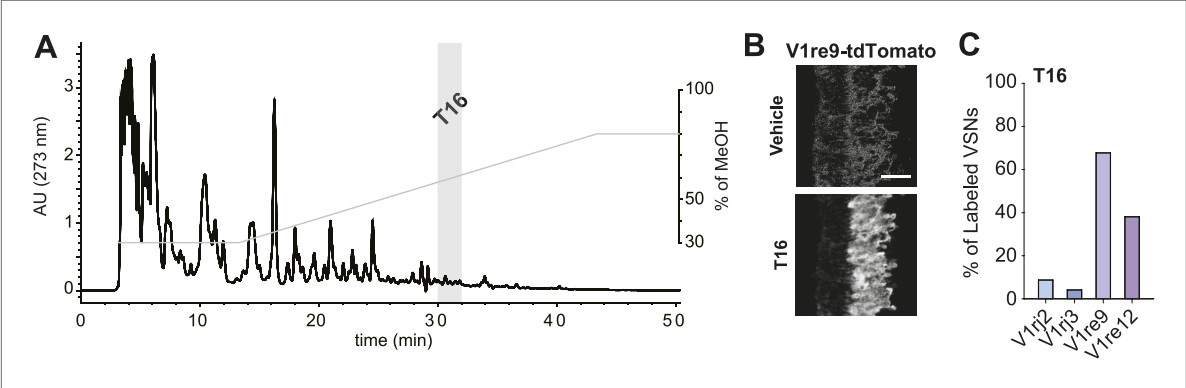

**Figure 7**. T16 fraction contains the female-specific gender cue. (**A**) Chromatogram of HPLC purification using a C18 column. The light gray bar indicates the T16 fraction that activates the V1re9 and V1re12 receptors. (**B**) A representative image of GCaMP2 VNO slices from a V1re9-tdTomato-expressing mouse responding to the T16 fraction. Scale bar, 50 µm. (**C**) Bar graph showing the percentage of V1rj2 (n = 172), V1rj3 (n = 243), V1re9 (n = 189) or V1re12 (n = 205) VSNs responding to the T16 fraction.

The following figure supplements are available for figure 7:

**Figure supplement 1**. Acetone fraction from XAD4 resins retains VNO-stimulatory activity.

## Discussion

### A novel strategy to identify functional pheromone receptors

Most mammalian pheromones have not been molecularly identified. Consequently, the majority of the vomeronasal receptors have not been characterized, nor have they been assigned with specific functions. In insect species, pheromones have traditionally been first identified through behavioral assays (*Wyatt, 2003*). The identification of specific receptors for the pheromones usually follows. The pairing of pheromones and their cognate receptors then leads to the characterization of the specific neural circuits involved in pheromone communication (*Dahanukar and Ray, 2011*; *Gomez-Diaz and Benton, 2013*). In mammals, chemical purification studies have revealed a number of urinary chemicals as putative pheromones (*Novotny et al., 1985*, *1986*; *Jemiolo et al., 1989*). However, few compounds are able to elicit behavioral or physiological responses by themselves. Moreover, some recent studies have raised doubts about the effects of some of these compounds as pheromones (*Flanagan et al.,*

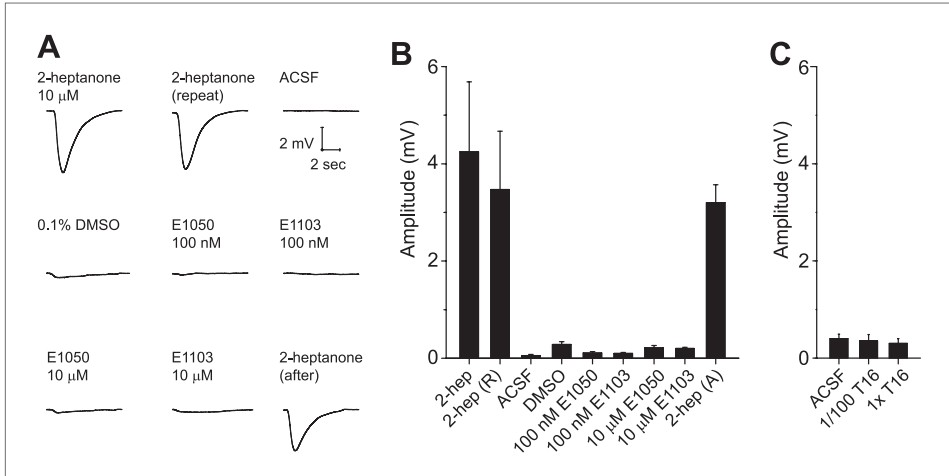

**Figure 8**. Sulfated estrogens and T16 fraction do not activate the main olfactory system. (**A**) Traces showing EOG responses to 2-heptanone, E1050 and E1103. (**B** and **C**) Bar graphs showing the mean amplitude of EOG responses to 2-heptanone, E1050 and E1103 (**B**) and the T16 fraction (**C**). Error bars: SEM (n = 3 mice).

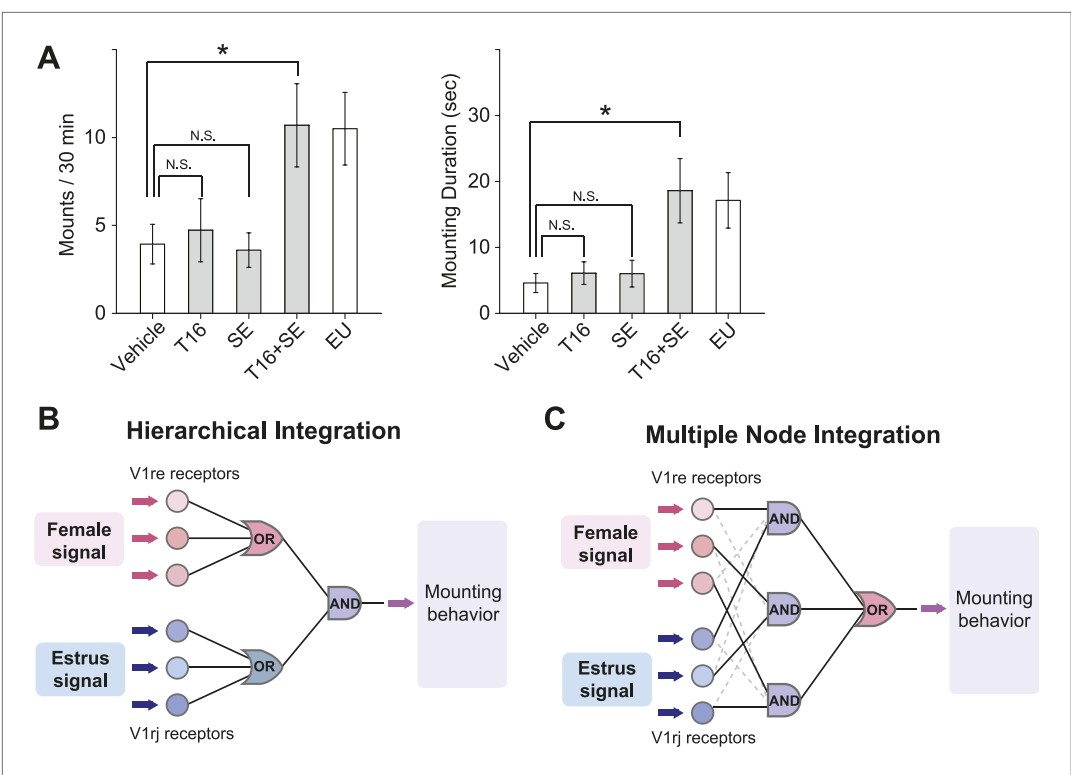

**Figure 9**. Combined female and estrus cues are sufficient to promote mounting behavior. (**A**) The number (left), duration (middle) and latency (right) of mounting behavior of naïve males toward females painted with vehicle, T16 (n = 13), SE (n = 10), T16+SE (n = 13) or EU samples. The data presented with white bars are the same as in **Figure 1**. Error bars: SEM; *p<0.05 (Mann–Whitney test). (**B** and **C**) Schematic illustrations of two alternative models of neural circuits that integrate the female and estrus signals.
The following figure supplements are available for figure 9:

**Figure supplement 1**. Sulfated estrogens promote courtship in conjunction with NEU.

**2011**; **Brechbuhl et al., 2013**; **Osada et al., 2013**). The lack of information about the ligands creates a conundrum in identifying functional receptor/ligand pairs and characterizing their downstream circuits.

We present here an approach to resolve this conundrum. Previously, we have established an approach to profile the response of individual VSNs to urine samples from both sexes with different genetic backgrounds or reproductive status such that the cells detecting specific signals can be identified even in the absence of specific information of pheromones (**He et al., 2008**). In this study, we extended this approach by combining it with single cell degenerate RT-PCR to clone and identify the receptor genes expressed in neurons that show specific response profiles.

In addition, we have developed a transgenic approach to characterize VRs and to purify their ligands. To date, there are only a few members of the V1r and V2r families of receptors that have been functionally expressed in heterologous expression systems (**Shirokova et al., 2008**; **Dey and Matsunami, 2011**). Adopting a transgenic approach not only eliminates the difficulty in establishing a robust heterologous assay system for each receptor, but also allows the analysis of receptor function in their native environment (**Zhao et al., 1998**; **Nguyen et al., 2007**; **Haga et al., 2010**). By taking advantage of the tetracycline-inducible system in transgenic mice, we ectopically expressed the identified V1r genes in the VNO to characterize their response properties. Moreover, the transgenic mice expressing the V1rs also enabled us to fractionate and partially purify their natural ligands. The physiological importance of these semi-purified ligands was confirmed by subsequent behavioral assays.

These approaches enable us to identify receptors that receive specific information whether or not their respective ligands are known. A similar approach has recently been established by using immediate early genes (IEGs) (**Isogai et al., 2011**). Double staining of IEGs and receptor genes by in situ

hybridization allowed the categorization of receptor groups based on their putative functions. However, conventional in situ hybridization techniques do not provide a clear distinction among highly homologous genes such as members of a given receptor clade. Moreover, in situ IEG staining is less sensitive than calcium imaging for measuring neuronal activation. These limitations can be overcome by using our current methodology.

## Ligand recognition by the vomeronasal receptors

The VSNs have been shown to be highly sensitive to their ligands (*Leinders-Zufall et al., 2000*, *2004*; *Kimoto et al., 2005*; *He et al., 2010*). Consistent with this, we found that the sulfated estrogen compounds activate the VSNs at sub-nanomolar concentrations: neurons can be activated by 100 pM concentrations of E1103 or E1050. Importantly, the same sensitivity was recapitulated by the ectopically expressed V1rj2 and V1rj3 receptors. The V1rj receptors are not only sensitive to sulfated steroids, but they are also highly selective. Except one sulfated androgen, none of the sulfated steroids other than estrogens activated the V1rj2 and V1rj3 receptors. On the other hand, although the V1rj2 and V1rj3 receptors are specifically activated by the sulfated estrogens, they tolerate some structural variation of the sulfated estrogens. The bisulfated estrogen, E1050, and the monosulfated estrogen, E1103, activated the V1rj receptors with comparable efficacy. These observations indicate that V1r receptors can discriminate distinct structural classes of steroids, while accepting minor structural variations within the class.

## Redundancy in female pheromone recognition

SEs and estrus signals are recognized by multiple members of V1rj group receptors while female-specific gender cues are recognized by several V1re-Chr.7 group receptors. This suggests that receptors that recognize a given female cue are closely related family members belonging to the same clade of the V1r family. Moreover, the members of each receptor group are clustered in the same chromosomal loci. The V1rj members are located in close proximity to each other on Chr.7 in a stretch of ~150 kb DNA. The receptors identified from FUSCs are V1re clade members also clustered on Chr.7. The V1re clade members located on Chr.17 were not identified from any of the FUSCs, despite their sequence similarity to those located on Chr.7. Although it is not clear whether the V1re-Chr.17 receptors also respond to female-specific gender cues, the evidence so far suggests a tight association between the chromosomal location of the receptor genes and their ligand recognition specificity. Thus, it is likely that the pools of receptor genes have been expanded locally in the genome to enable the redundant recognition of estrus or gender specific signals. This expansion may also diversify the response properties of the receptor pools. V1rj3 appears to be narrowly tuned to sulfated estrogens and is more sensitive to E1103. V1rj2, on the other hand, is more sensitive to E1050. In addition, it can be activated by additional sulfated steroid compounds. These diversified response properties may allow the fine tuning of behavioral response.

Similar to our findings, other studies also showed the recognition of a given ligand by multiple receptors. For example, one of the ESP family peptides (*Kimoto et al., 2007*), ESP5, is recognized by both V2rp1 (Vmn2r112) and V2rp2 (Vmn2r111), both located on Chr.17 (*Dey and Matsunami, 2011*). In another case, the MHC peptide 'AAPDNRETF' activates not only V2r1b (Vmn2r26)-positive but also V2r1b-negative VSNs (*Leinders-Zufall et al., 2009*). 2-heptanone stimulates V1rb2-expressing VSNs (*Boschat et al., 2002*), but it can still stimulate the VNO of the mutant mice with a genomic deletion of the V1ra and b clusters (*Del Punta et al., 2002*). The identity of the additional receptors, however, remains unknown in the latter two cases.

It should be noted that some pheromones are specifically recognized by a single cognate vomeronasal receptor. For instance, the male specific ESP family peptide, ESP1, is recognized by V2rp5, and the deletion of V2rp5 completely abolishes VNO activation and the behavioral response to ESP1 (*Haga et al., 2010*). The highly specific pairing between ESP1 and V2rp5 enables a labeled line transmission of ESP1 information in triggering lordosis behavior. However, a highly specific labeled line carries a price—the disruption of a single receptor gene completely eliminates the behavioral effect of ESP1.

The observation that several receptors recognize the same set of female pheromone cues suggests that the receptors may serve a redundant function in encoding pheromone information. Building a redundancy in pheromone recognition would enable robust detection of important cues. Both the estrus and the female-specific gender cues are recognized by multiple receptors, and both are required to trigger male mounting behaviors. Since male mounting behavior is an essential component of male

reproductive behavior, redundant coding of these signals may be advantageous to reproductive success. In addition, activation of multiple receptors by a given ligand may offer an advantage in detecting a broader range of ligand concentrations. Indeed, in the present study, the SEs showed different threshold concentrations in activating V1rj2 and V1rj3. Such diversity should enable the VNO to detect the threshold concentration of the ligand with the most sensitive receptor, as well as to accommodate high concentrations of the ligand that can fully activate several receptors. In this scenario, redundancy offers an advantage in expanding the dynamic range of pheromone detection and providing quantitative information of a given pheromone ligand.

## Plausible logic in pheromone information processing

VNO activation triggers endocrine changes and ritualistic action patterns through preprogrammed neural circuits. Individual pheromones can directly trigger distinct innate behaviors, but complex interactions of pheromones in eliciting behaviors are observed in vertebrate species (*Wyatt, 2003*). Our observations suggest a model of neural circuitry that processes complex pheromone cues. In this model, activation of both the V1rj and V1re-Chr.7 receptor circuits promotes mating behavior in male mice. This requirement of two sets of pheromone cues suggests an 'AND' gate in circuit logic, which can provide safeguard against spurious signals. The gender cue is not sufficient to induce mounting behavior unless the estrus signal is also present. This mechanism may enhance reproductive success by inducing males to mate preferentially with estrous females. Likewise, the activation of the V1rj receptors alone is not sufficient to induce mounting behaviors. Estrogen metabolites, including sulfated estrogens, may be excreted by animals of other species. Responding to these compounds alone may induce an inappropriate behavioral response to false targets. It is conceivable that only when both estrus signals and gender specific signals from their own species are present, the mating circuit is activated. On the other hand, the observation that several receptors recognize the same set of female pheromone cues suggests that the receptors may serve a redundant function in encoding pheromone information, which resembles an 'OR' gate in circuit logic. Building a redundancy in pheromone recognition would enable robust detection of important cues with expanded dynamic range and provide better quantitative information of a given pheromone ligand.

An intriguing question is where the signal integration occurs in the brain. One potential site is the connection between the VSN axon terminals and dendrites of mitral cells within the glomeruli in the accessory olfactory bulb (AOB), the primary relay center in the vomeronasal system. It has been shown that VSN axons expressing homologous receptors intermingle within spatially conserved domains of the AOB (*Wagner et al., 2006*), providing a potential integration site for 'OR' gated signals. The AOB secondary neurons, mitral cells, arborize multiple dendrites onto different glomeruli (*Takami and Graziadei, 1991*), providing an anatomical structure for a potential integration site of 'AND' gated signals. This scenario fits to the hierarchical integration model (*Figure 9B*), where the female or estrus signals received by multiple members of V1re or V1rj receptors reach the 'OR' gates for each receptor group, and then the signals passing through the gates are integrated again at the 'AND' gate. Alternatively, it is also possible that the female signals received by any of V1re receptors are integrated with the estrus signals received by any of V1rj receptors at the multiple nodes of the 'AND' gates. The signals that passed through each 'AND' gate are integrated again at the 'OR' gate (multiple node integration model, *Figure 9C*). These integration events may also occur at higher-level circuitry in the brain. Anatomical tracing of the circuits will further reveal this logic.

In conclusion, our results suggest that the VNO system may harness these two opposing logical mechanisms to fulfill divergent duties in pheromone signal processing: detecting cues with high sensitivity while preventing non-specific induction of behaviors by false signals. This logic complements the highly specific, 'labeled line' circuit that directly links sensory stimuli to behavioral responses, such as the one that regulates lordosis behaviors triggered by the peptide ESP1 (*Haga et al., 2010*).

# Materials and methods

## Mice

All mice were maintained in Lab Animal Service Facility of Stowers Institute at 12:12 light/dark cycle and provided with food and water *ad libitum*. Experimental protocols were approved by the Institutional Animal Care and Use Committee at Stowers Institute (Protocol 2013-0117) and were in compliance with NIH Guide for Care and Use of Animals. TetO-GCaMP2 (*He et al., 2008*) mice and OMP-IRES-tTA

(OIVT) (*Yu et al., 2004*) mice were described previously. These two lines were crossed to induce GCaMP2 expression in the vomeronasal and olfactory sensory neurons. Responses to urine and other chemical stimuli were recorded from the VNO isolated from OIVT; tetO-GCaMP2 mice at 2-6 months of age. To generate the tetO-V1r-IRES-tdTomato lines, coding region of either V1re9, V1re12, V1rj2 or V1rj3 was cloned from the genomic DNA of C57BL/6 mice by PCR and placed in between the tetracycline-dependent promoter (tetO) and IRES-tdTomato. The DNA was then linearized and injected to the pronuclei of zygotes from F1 (CBA/J; C57Bl/10J) mice. Founders were crossed to OIVT and Gγ8-tTA (*Nguyen et al., 2007*) to generate compound heterozygotes containing OIVT, Gγ8-tTA and tetO-V1r-IRES-tdTomato alleles so that ectopic V1rs are expressed in the VNO. These mice are further crossed to TetO-GCaMP2 mice to generate compound heterozygotes containing all four alleles.

Sexually naïve 3-month-old male C57BL/6J mice were used to examine courtship behaviors. Overiectomized females were prepared using 2-month-old BALB/c females by enucleating the ovaries. These females were used as recipient females at 3–6 months of age. To collect urine samples from females at different estrous status, 3- to 4-week-old female C57BL/6 mice were injected with pregnant mare's serum gonadotropin (PMSG) to induce synchronized estrus.

## Urine samples and chemicals

Urine samples were collected from 5 IU of PMSG-injected females from 1 day before to 5 days after injection in metabolic cages. Urine collected 2–3 days after PMSG injection were considered as estrus samples. Non-estrus samples were collected 4–5 days after PMSG injection. The freshly collected urine samples were centrifuged to remove solid contaminants and frozen at −80°C until use. For calcium imaging experiments, each urine sample was diluted 100 times with Ringer's solution (in mM: 125 NaCl, 2.5 KCl, 2 $CaCl_2$, 2 $MgCl_2$, 25 $NaHCO_3$, 10 HEPES and 10 glucose). For behavioral experiments, each urine sample was painted to recipient females without dilution.

Sulfated steroids were purchased from Steraloids (Newport, RI), the catalog IDs were used to label the compounds. The sulfated steroids used in this study are summarized in the *Supplementary file 3*. Each steroid was diluted with dimethyl sulfoxide (DMSO) to generate 20 mM stock solution before further diluted with Ringer's solution to the concentrations tested.

## Calcium imaging with VNO slices

Details of imaging setup and procedures were described previously (*Ma et al., 2011*). Briefly, VNO slices were maintained in carboxygenated (95% $O_2$, 5% $CO_2$) mouse artificial cerebrospinal fluid (mACSF; in mM: 125 NaCl, 2.5 KCl, 1 $MgCl_2$, 2 $CaCl_2$, 1.25 $NaH_2PO_4$, 25 $NaHCO_3$ and 10 glucose) at room temperature. Carboxygenated mACSF was also used to superfuse VNO slices at a speed of 1 ml/min. The flow was kept unidirectional by placing the inlet and outlet at the apical and basal sides of VNO epithelium, respectively. Urine samples and steroids were delivered through the HPLC injection valve in Ringer's solution. To minimize mechanical artifacts, a continuous flow (~0.3 ml/min) of Ringer's solution was maintained during the experiment. Solutions were switched by using the injection valve without disrupting the flow.

Time-lapse acquisition of GCaMP2 signals from a VNO slice was performed on AxioScope FS2 (Carl Zeiss, Oberkochen, Germany) microscope with a 20X/0.5NA water-dipping lens. In experiments using OMP-IRES-tTA; tetO-GCaMP2 mice, GCaMP2 was excited at 930 nm (Ti-Sapphire laser, Coherent, Santa Clara, CA) and visualized through a 685 nm short-pass emission filter. In the experiment of Gg8-tTA; OMP-IRES-tTA; tetO-V1r-tdTomato; tetO-GCaMP2 mice, TdTomato and GCaMP2 were excited at 950 nm and visualized through 560 nm long-pass and 500–550 band-pass filters, respectively.

Image processing and data analysis, including region of interest (ROI) detection and automated signal analyses, were performed using computer programs written in Matlab. For ROI detection, VNO images for each stimulus were first registered using the elastic registration library function of Axovision (Carl Zeiss), with the first frame chosen as reference. Then, for each stimulus, the local (specific to the stimulus) ROIs were detected in two steps: (1) generate ROI candidates by using a customized signal-to-noise ratio criteria; (2) select the candidates as ROIs with manual inspection. Subsequently, the local ROIs were integrated to generate a list of global (non-specific to any stimulus) ROIs. For signal peak analysis, the temporal profiles of image intensity at the global ROIs were individually extracted as raw signals and a scalar value ΔF/F was calculated. In order to compute F, a baseline fitting step was performed to model photo-bleaching effect and a peak detection step was performed to model the signal peak. ΔF was computed as the difference between the signal peak and the baseline.

The computation was then manually validated to exclude possible errors. A threshold of 30% ΔF/F was imposed to identify VSNs as responding to a stimulus.

## Receptor cloning

To isolate VSNs exhibiting a specific response profile, calcium imaging was conducted with a 40X/0.8NA water-dipping lens. GCaMP2 signal was excited by EXFO X-Cite 120PC light source equipped with a band-pass filter (450–490 nm). The epifluorescent images were acquired by a CCD camera (Carl Zeiss HRM). Under bright-field and superimposed fluorescent illumination, the cytoplasmic content of an identified VSN was aspirated into a glass capillary (2 µm tip size). The content was transferred to a PCR tube containing lysis buffer (*Kurimoto et al., 2007*). Samples were immediately frozen on dry ice and kept at −80°C until use.

cDNAs of each sample was reverse transcribed from total RNA and amplified by the procedure described previously (*Kurimoto et al., 2007*). First-strand cDNAs were synthesized using a V1 (dT)$_{24}$ primer. Unreacted primer was specifically eliminated by exonuclease treatment and second strands were then generated with a V3 (dT)$_{24}$ primer after poly(dA) tailing of the first-strand cDNAs. By again using V1 (dT)$_{24}$ and V3 (dT)$_{24}$ primers, the double-stranded cDNAs were then amplified by 20-cycle-PCR. V1r or V2r genes were amplified by 17 V1r and 7 V2r degenerate PCR primers (*Supplementary file 1*). Each primer pair was examined for the coverage of members of each receptor group by performing RT-PCR analysis by using either pooled VNO mRNA samples or single cell samples (*Supplementary file 2*). PCR products were cloned into pGEM-T vector followed by sequence analysis. The sequence annotation was conducted by using the custom-made program with the custom-made receptor database containing all identified VRs.

## In situ hybridization

In situ hybridization is performed following the protocol described by *Ishii et al. (2004)*.

## Urine purification

4 ml of urine was incubated with 2 g of a polymeric adsorbent resin, Amberlite XAD4 (Sigma-Aldrich, St. Louis, MO), overnight at 4°C. After incubation, the mixture of urine and the resin was transferred to a poly-prep chromatography column (Bio-rad, Hercules, CA) and the flow-through was gravitationally collected. 35 ml of ultrapure water and the same amount of acetone were sequentially applied to the resin, and eluted fractions were collected, lyophilized and stored at −20°C until use.

The acetone fraction was dissolved in 30% methanol (MeOH), 0.1% trifluoroacetic acid (TFA) in water and passed through a 0.2 µm PTFE membrane filter (Sigma-Aldrich). The filtrated sample was then loaded onto a reverse-phase high-performance liquid chromatography (HPLC) column (Atlantis T3, 10 mm × 250 mm, Waters, Milford, MA). Bound compounds were eluted with a gradient of 30–80% MeOH, 0.1% TFA in water at 4 ml/min, and eluted fractions were collected every 2 min (fraction T1-25). Fractions were lyophilized and stored at −20°C until use.

For VNO imaging experiment, lyophilized fractions were dissolved in Ringer's solution with final concentration correspond to 1/10 of the original urine sample used. For behavioral experiment, lyophilized T16 fraction was dissolved in PBS with the final concentration was equivalent to the original urine sample used.

## Electro-olfactogram (EOG) recordings

Field potentials were recorded from the ciliated layer of intact MOE. Briefly, the MOE was exposed and perfused with oxygenated mACSF. Field potential was recorded using glass pipettes (10 µm diameter) connected to an AI 401 pre-amplifier (Molecular Devices, Sunnyvale, CA). The signals were further amplified by a signal conditioner (Molecular Devices), digitized at 1 kHz by Digidata 1322A (Molecular Devices), low-pass filtered at 20 Hz and further analyzed using pCLAMP. Stimuli were delivered through a second glass pipette controlled by the HPLC injection valve. Applications of ACSF and 0.1% DMSO were used for evaluating the baseline activities. Multiple applications of 2-heptanone (twice at a beginning and once at the last) was used for evaluating the tissue viability.

## Behavior assay

Individual males were single-housed for at least 3 weeks without bedding change prior to the test. Assays were performed in the home cages. Males were habituated to the behavioral room by transferring

their cages to the room for 1 hr each day for 3 days prior to the assay. On the day of assay, males were again habituated for 30 min before recipient females were introduced into male cages. The females were painted with one of the following stimuli: vehicle (PBS), estrus or non-estrus urine, 10 µM sulfated estrogens (E1050, E1103) in vehicle or in none-estrus urine, HPLC fraction in vehicle or with 10 µM sulfated estrogens. The stimuli were painted on the anogenital area and dried for 10 min before the females were introduced to male home cages. All assays were conducted in the dark cycle of the animals and video-taped. To minimize the influence of behavioral differences of the recipient females, each female was randomly assigned to each of the stimuli and used multiple times with at least 2 weeks of interval. For aggression assay, individual males were singly housed for 2–3 weeks. During the test, a group-housed CBA male was introduced to the home cage of the resident male.

All the behavioral data were scored manually using Observer XT software (Noldus Information Technology, Wageningen, Netherlands). Male mounting behavior was counted based on the definitions as follows: (1) males approach females and run his head first along her flank then onto the back, (2) male arms are locked in front over female's hips and over her middle section so that males and females are positioned genital to genital, (3) males show the pelvic thrust with a stable frequency. At least 10 naïve males were tested with each stimulus. Male aggression was scored when the resident male attacked and bit the flank of the intruder male. Kruskal–Wallis test and Mann–Whitney's post-hoc test were used to analyze the data on the frequency, duration and latency of mounting behavior.

## Acknowledgements

We thank Drs R Krumlauf, R Axel, JW Wang, K Touhara and N Yamanaka for helpful discussion and comments on the manuscript. We appreciate the generosity of Dr N Ryba for providing Gγ8-tTA mice. We are grateful for technical assistance from J Xin, S Loesgen, V Ramalingam, D Lee, K Jensen, A Satterlee, A Moran, M Durnin, QE Yu, R Alexander, M Richardson, A Paulson, Y Zhang, D Bradford and the Lab Animal Services, Microscopy Center, Proteomics and Molecular Biology Core Facilities at the Stowers Institute. This work is supported by funding from the Stowers Institute and the US National Institutes of Health (National Institute on Deafness and Other Communication Disorders 008003) to CRY. SH-Y was supported by a postdoctoral fellowship from the Japan Society for the Promotion of Science.

## Additional information

### Funding

| Funder | Grant reference number | Author |
| --- | --- | --- |
| Stowers Institute for Medical Research | | C Ron Yu |
| National Institutes of Health | R01DC008003 | C Ron Yu |
| Howard Hughes Medical Institute | | Luke D Lavis, Loren L Looger |
| Japan Society for the Promotion of Science | | Sachiko Haga-Yamanaka |

The funders had no role in study design, data collection and interpretation, or the decision to submit the work for publication.

### Author contributions

SH-Y, LM, Conception and design, Acquisition of data, Analysis and interpretation of data, Drafting or revising the article, Contributed unpublished essential data or reagents; JH, QQ, Acquisition of data, Analysis and interpretation of data; LDL, Conception and design, Contributed unpublished essential data or reagents; LLL, Conception and design, Drafting or revising the article; CRY, Conception and design, Analysis and interpretation of data, Drafting or revising the article

### Ethics

Animal experimentation: Experimental protocols were approved by the Institutional Animal Care and Use Committee at Stowers Institute (Protocol 2013-0117) and were in compliance with NIH Guide for Care and Use of Animals.

## Additional files

### Supplementary files

• Supplementary file 1. Summary of degenerate PCR primers. List of primers designed against different clade members of the V1r and V2r families of genes.

• Supplementary file 2. Coverage of degenerate primers. Receptor genes identified from degenerate RT-PCR using whole VNO and single cells. Results from single cells are pooled from multiple experiments. In the Coverage column, bold face numbers indicate 100% coverage of the clade members and italicized numbers indicate coverage lower than 50%.

• Supplementary file 3. Sulfated steroids tested. List of the Steraloids IDs and names of sulfated steroids tested in the experiments.

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
