## [Decision Letter]

Thank you for sending your work entitled “Integrated Action of Pheromone Signals in Promoting Courtship Behavior in Male Mice” for consideration at *eLife*. Your manuscript has been evaluated by three expert reviewers and a member of the Board of Reviewing Editors (Jeremy Nathans), all of whom are enthusiastic about it. The following comments reflect the consensus view of these four individuals.

The Reviewing editor and the other reviewers discussed their comments before we reached this decision, and the Reviewing editor has assembled the following comments to help you prepare a revised submission.

This is an impressive and important manuscript describing different components in female urine that activate VNO neurons expressing specific VR clades in males. Using a technically difficult combination of slice calcium imaging and single-cell RT-PCR, the authors identified two subtypes of VRs, V1rjs that respond to sulfated estrogen products (found only in urine from females in estrous) and V1res that respond to a specific fraction of female urine (T16). Using transgenic ectopic overexpression the authors demonstrate that ectopic expression of these receptors confers responsiveness to VNO neurons that normally do not express these receptors, supporting their hypothesis that these are the cognate receptors for these ligands.

The dissection of the genes that are likely to contribute and the chemicals that stimulate mating responses is an important new finding. A very significant aspect of the manuscript is the convincing evidence regarding the logic of the mechanisms that build the signals required to elicit behavioral responses. Although additional evidence will be needed to demonstrate genetic sufficiency of the identified receptors, the present work considerably focuses that effort.

Major points (points 1, 2, and 5 are the most significant; points 3 and 4 are of interest but are not as central):

1) The use of degenerate PCR is obviously essential to the authors' approach, but there is insufficient information to evaluate whether the receptors identified reflect PCR bias (positively or negatively). For example does one see a broad repertoire if random cells are chosen (or if VR are amplified from a bulk pool of mRNA)? Additionally, can V1rj1 be identified with the receptor oligos? This point is critical to the subsequent assignment of these receptors as detectors of the cognate ligands. One approach to this question would be to PCR amplify total VNO cDNA with the degenerate primers, subject the resulting mixture of PCR products to NextGen sequencing, and then use the read counts to quantitatively assess the repertoire that is revealed by PCR. This would allow one to recast the statement in the Results section: “It is highly improbably that, out of more than 300 VRs, the receptors belonging...” as a quantitative statement.

2) It is not clear how many cells were ’profiled‘ in order to get the 15 cells that expressed the restricted subset of VRs. If the authors looked at only 15 and got at least 1 receptor from each of them that would support the conclusion that the repertoire of receptors that can respond is restricted. Therefore it would be helpful to indicate that the data came from isolating the contents of X cells from which Y expressed VRs (where X and Y indicate the numbers).

3) Showing a causal relationship between activation of V1R e and j members and mating behavior is beyond the scope of the current study, as this would require knocking out these genes. That said, in odorant receptor M71 overexpressing mice, the mice show impaired detection of the M71 ligand acetophenone. Have the authors checked if any of the V1R overexpressing males show normal mounting behavior? It would be interesting to see how the transgenic mice that express V1rj and V1re in most cells of the VNO respond behaviorally to total female urine or SE and T16. Given that these receptors are now expressed in VNO neurons involved in other functions (for example aggression) one would expect that these mice might show some other behavioral alterations.

4) Could there also be female-specific pheromones that are still produced in ovarectomized females? In other words, if the authors paint a castrated male with the two SEs and T16 would males attempt to mate with them?

5) The authors do not make any reference to expression of the VNO receptors in females. Are they expressed in similar frequency as in the male VNO, and if so do they respond to the same compounds?

---

## [Author Response]

*1) The use of degenerate PCR is obviously essential to the authors' approach, but there is insufficient information to evaluate whether the receptors identified reflect PCR bias (positively or negatively). For example does one see a broad repertoire if random cells are chosen (or if VR are amplified from a bulk pool of mRNA)? Additionally, can V1rj1 be identified with the receptor oligos? This point is critical to the subsequent assignment of these receptors as detectors of the cognate ligands. One approach to this question would be to PCR amplify total VNO cDNA with the degenerate primers, subject the resulting mixture of PCR products to NextGen sequencing, and then use the read counts to quantitatively assess the repertoire that is revealed by PCR. This would allow one to recast the statement in the Results section: “It is highly improbably that, out of more than 300 VRs, the receptors belonging...” as a quantitative statement*.

Prior to using the degenerate primers to identify the female pheromone receptors, we tested and analyzed the coverage of these degenerate primers in amplifying the VR genes. Using cDNA from the whole VNO, we were able to identify majority of known members of the V1r and V2r families from every clade. A table is added to illustrate the result. In addition to the whole VNO samples, we have results from many single cells from control and experiments designed to identify receptors for specific ligands. Results from single cells indicated that some receptors were more frequently identified than others. This observation may be related to the difference in abundance. The results are shown in [Supplementary-material SD2-data].

V1rj1 is amplified. In fact, it is one of the receptors we identified from neurons responding to SEs, which is identical to Vmn1r86. The sequence is annotated in NCBI database (http://www.ncbi.nlm.nih.gov/nuccore/AY044668.1). However, perhaps due to the imperfection of the DNA sequence submitted to NCBI, V1rj1 is not annotated in Blat and Ensemble. Moreover, for some reason it is not linked with Vmn1r86 in NCBI: http://www.ncbi.nlm.nih.gov/nuccore/NM_001167536.1). To clarify this point, we have indicated this fact in a note.

*2) It is not clear how many cells were ’profiled‘ in order to get the 15 cells that expressed the restricted subset of VRs. If the authors looked at only 15 and got at least 1 receptor from each of them that would support the conclusion that the repertoire of receptors that can respond is restricted. Therefore it would be helpful to indicate that the data came from isolating the contents of X cells from which Y expressed VRs (where X and Y indicate the numbers)*.

The result was obtained from 6 animals. Approximately 50 cells in each slice responded to SEs. We picked 22 SE-responding cells and subjected them to degenerate RT-PCR. 15 of these cells were chosen for further analyses based on the criterion that the sample was not amplified by more than 3 pairs of degenerate PCR primers. Although the primer pairs were designed against specific sets of receptor genes, the same product could be amplified by more than one pair of primers. Out of the 15 cells analyzed, 14 cells of them expressed V1rj members.

*3) Showing a causal relationship between activation of V1R e and j members and mating behavior is beyond the scope of the current study, as this would require knocking out these genes. That said, in odorant receptor M71 overexpressing mice, the mice show impaired detection of the M71 ligand acetophenone. Have the authors checked if any of the V1R overexpressing males show normal mounting behavior? It would be interesting to see how the transgenic mice that express V1rj and V1re in most cells of the VNO respond behaviorally to total female urine or SE and T16. Given that these receptors are now expressed in VNO neurons involved in other functions (for example aggression) one would expect that these mice might show some other behavioral alterations*.

This is an interesting question. Our preliminary results indicate that the male mice expressing an ectopic receptor (OMP-IRES-tTA:tetO-V1rj3) exhibit relatively normal mating behaviors but diminished territorial aggression.

These observations are consistent with previous studies showing that VNO is required for pheromone-triggered territorial aggression. The interpretation of this result, however, is complicated by many factors. The expression of ectopic receptor may displace the endogenous receptor genes to effect a general reduction of VNO function. Alternatively, it is possible that suppressed expression of VRs responding the male cues selectively reduces male aggression. Further complicating the interpretation is that the main olfactory system may be compromised as well. Similar to the ectopic expression of M71 receptor in the monoclonal nose mice, our transgenes are also driven by OMP, which allows the expression of VRs in the main olfactory epithelium. A systematic investigation of different lines will be required to address the questions being raised.

*4) Could there also be female-specific pheromones that are still produced in ovarectomized females? In other words, if the authors paint a castrated male with the two SEs and T16 would males attempt to mate with them*?

Urine samples from ovariectomized females do activate V1re9/12 expressing cells. However, we found that these receptors were activated to lesser extent by urine from ovariectomized females. The data is presented in Figure 5—figure supplement 3. The result suggests that the female identity cue is partially dependent on an intact ovary.

*5) The authors do not make any reference to expression of the VNO receptors in females. Are they expressed in similar frequency as in the male VNO, and if so do they respond to the same compounds*?

We have conducted in situ hybridization experiments as well as imaging experiments using VNO from both male and female animals. In situ data show that the expression patterns of the receptor genes are not sexually dimorphic. Moreover, VNO slices from male and female mice show similar responses to the SEs and estrous urine. This is an important point and we have added a figure to illustrate it (revised Figure 6).